# Global response of parameterised convective cloud fields to anthropogenic aerosol forcing

Zak Kipling[1,*], Laurent Labbouz[1,2], and Philip Stier[1]

[1]Atmospheric, Oceanic and Planetary Physics, Department of Physics, University of Oxford, Oxford, UK
[2]Laboratoire d'Aérologie, Université Paul Sabatier, Toulouse, France.
[*]now at: European Centre for Medium-Range Weather Forecasts, Reading, UK

**Correspondence:** Zak Kipling (zak.kipling@ecmwf.int)

**Abstract.** The interactions between aerosols and convective clouds represent some of the greatest uncertainties in the climate impact of aerosols in the atmosphere. A wide variety of mechanisms have been proposed by which aerosols may invigorate, suppress, or change the properties of individual convective clouds, some of which can be reproduced in high-resolution limited-area models. However, there may also be mesoscale, regional or global adjustments which modulate or dampen such impacts which cannot be captured in the limited domain of such models. The Convective Cloud Field Model (CCFM) provides a mechanism to simulate a population of convective clouds, complete with microphysics and interactions between clouds, within each grid column at resolutions used for global climate modelling, so that a representation of the microphysical aerosol response within each parameterised cloud type is possible.

Using CCFM within the global aerosol–climate model ECHAM–HAM, we demonstrate how the parameterised cloud field responds to the present-day anthropogenic aerosol perturbation in different regions. In particular, we show that in regions with strongly-forced deep convection and/or significant aerosol effects via large-scale processes, the changes in the convective cloud field due to microphysical effects is rather small; however in a more weakly-forced regime such as the Caribbean, where large-scale aerosol effects are small, a signature of convective invigoration does become apparent.

## 1 Introduction

The indirect effects of atmospheric aerosol via interactions with cloud and precipitation remain some of the most uncertain contributors to anthropogenic radiative forcing of the earth's climate (Boucher et al., 2013; Myhre et al., 2013). Although the principle behind the cloud albedo effect (Twomey, 1977) is relatively well understood, its magnitude is much less certain, while potential effects on cloud cover and liquid water path and semi-direct effects (Hansen et al., 1997; Ackerman et al., 2000) are more tentative. For convective and mixed-phase clouds, a number of further effects have been proposed relating to enhanced or retarded glaciation and consequent changes in latent heat release (Lohmann and Feichter, 2005; Rosenfeld et al., 2008); Fan et al. (2013) also propose effects based on changes in ice-particle fall speed. Aside from effects on the radiation balance, aerosol–cloud interactions have the potential to alter the distribution of precipitation. Although total precipitation is constrained to balance evaporation and by energetic constraints (Mitchell et al., 1987; Allen and Ingram, 2002), changes in

location and frequency of different intensities are possible, which may have significant consequences for drought and flood

risk (e.g. Muller and O'Gorman, 2011; Hodnebrog et al., 2016; Myhre et al., 2016).

However, although these mechanisms may be captured in idealised cloud-resolving models (CRMs) and large-eddy simulations (LES) of individual clouds (e.g. Tao et al., 2007; Lee, 2012), there is little robust observational evidence, and it has been suggested that such effects are countered by negative feedbacks or buffering processes on larger spatio-temporal scales (Stevens and Feingold, 2009). Furthermore, evidence from observations may suffer from a lack of significance or correlation

between aerosol and the thermodynamics to which convection is much more sensitive (Varble, 2018), and that from CRM and LES studies from uncertainty in the complexities of microphysical parameterisation (White et al., 2017). Many of these topics are covered in more detail in the recent review by Fan et al. (2016).

Studying the effects of aerosol on convection using models at the global scale is challenging, however, as we are some way from having the computer power to routinely run convection-resolving global climate models for more than short time peri-

ods. Current models therefore all require some form of sub-grid-scale parameterisation of sub-grid-scale physical processes, including convection (e.g. Jakob, 2010). The convective parameterisations used in most current models are of the bulk mass flux variety, representing sub-grid-scale convection by one "average" convective cloud, either with an estimation of the vertical updraught velocity (e.g. Kim and Kang, 2012), or more traditionally without any explicit separation of the total updraught mass flux into horizontal area and vertical velocity (e.g. Tiedtke, 1989; Gregory and Rowntree, 1990; Bechtold et al., 2001).

Such schemes are very effective for many purposes, but are unable to represent either the heterogeneity of convective clouds within a grid column, or the detailed microphysics through which indirect aerosol effects may operate (especially the activation process which depends on vertical velocity and supersaturation and is highly non-linear). This is one of the reasons why, although global models may represent the global distribution of *mean* precipitation quite well, both the diurnal cycle and intensity distribution of convective precipitation remain a challenge (Stratton and Stirling, 2012; Kooperman et al., 2018). Current

model-based estimates of the effective radiative forcing due to aerosol–cloud interactions (ERFaci) may also be limited by the fact that the models mostly represent only the interactions with large-scale stratiform clouds. These interactions are typically represented by coupling prognostic size-resolved cloud microphysics and aerosol schemes via a parameterisation of the aerosol activation process by which some aerosols act as cloud condensation nuclei (CCN). The activation itself, and thus potentially the overall contribution of ERFaci to climate sensitivity, depends crucially on local vertical velocity and its variability (West

et al., 2014; Donner et al., 2016), which as noted above is indeterminate in the convective context for many schemes.

There have been a number of previous attempts to represent aerosol–convection interactions in parameterised convection by a variety of approaches. Nober et al. (2003) used in-situ observed or satellite-retrieved cloud droplet number concentration (CDNC) to modulate the simple microphysics within a Tiedtke-based bulk mass flux scheme. Menon and Rotstayn (2006) introduce simple parameterisations of CDNC or droplet size based on prognostic aerosol into two different GCMs using quite

different cloud and convection schemes. Lohmann (2008) goes further, introducing full two-moment microphysics into bulk mass-flux convection, driven by a parameterisation of aerosol activation notwithstanding the lack of explicit information about the convective vertical velocity or its variation amongst updraughts. Another two-moment microphysics parameterisation has been implemented within the Zhang and McFarlane (1995) convection scheme (Song and Zhang, 2011; Song et al., 2012),

using a plume model to derive the convective vertical velocity and aerosol activation, but still without representing the diversity of convective clouds (and hence of vertical velocities) that may exist within one model grid cell. Other approaches for parameterising convection include unified schemes which aim to treat all cloud types along with turbulence in a single set of equations: for example CLUBB (Thayer-Calder et al., 2015) applies such an approach in multiple sub-columns to represent a variety sub-grid-scale eddies and clouds. The convective scheme in the GFDL AM3 model (Donner, 1993; Donner et al., 2001, 2011) attempts to parameterise some mesoscale structure in addition to the updraughts and downdraughts, using cloud distributions derived from observations. Finally, super-parameterisation uses an embedded cloud resolving model (typically 2D) inside each column of the host model – this approach is well suited to representing aerosol–cloud interactions (e.g. Wang et al., 2011), but at a very high computational cost and still lacking non-local interactions (i.e. the ability for convective elements at the edge of one host-model column to interact with those at the edge of the neighbouring column other than via the large-scale dynamics).

The Convective Cloud Field Model (CCFM), introduced by Nober and Graf (2005) and developed further by Wagner and Graf (2010), provides another approach, representing a spectrum of different convective clouds within the grid cell, with the number of each cloud type dynamically determined by the environment. Each cloud type is represented by a separate realisation of a physical cloud model, with its own horizontal area, vertical velocity and in-cloud microphysics. In Kipling et al. (2017), CCFM was further extended with a sub-cloud dry convection model to describe the triggering of moist convection and determine the cloud-base vertical velocity crucial for aerosol activation, and evaluated in the global model ECHAM6–HAM2 both against observations, performing comparably to the standard Tiedtke–Nordeng bulk mass-flux convection scheme used in that model and showing an improvement in the diurnal cycle of convection in certain regions. The CCFM cloud spectra themselves have been evaluated against observations by Labbouz et al. (2018).

We use CCFM here to study the impact of anthropogenic aerosol emissions on convective clouds in a model which can represent both local variations in the convective cloud field and feedbacks via the global atmosphere (though still not those via the ocean, without moving to a coupled atmosphere–ocean GCM). In this way, we quantify the responses of the parameterised convective cloud fields in several regions to changing aerosol, and ask which physical processes are dominating these responses.

## 2 Model description

### 2.1 The ECHAM–HAMMOZ global composition–climate model

The modelling framework used here is as in Kipling et al. (2017); further details can be found there but we repeat the main points here for clarity. ECHAM–HAMMOZ is built upon the global atmospheric general circulation model ECHAM6.1 (Roeckner et al., 2003; Stevens et al., 2013), coupled to the interactive aerosol module HAM2.2 (Stier et al., 2005; Zhang et al., 2012). The system also includes a gas-phase chemistry model (the "MOZ" part), but this is not active in the configuration used here which includes only a simplified sulfate precursor scheme driven by DMS and $SO_2$ emissions which drives condensation and new particle formation.

Developed at the Max Planck Institute for Meteorology, ECHAM6 (Roeckner et al., 2003; Stevens et al., 2013) is based around a spectral dynamical core using a vorticity–divergence formulation. In the vertical, a hybrid sigma/pressure coordinate is used. The physical parameterisations are implemented on a Gaussian grid corresponding to the spectral truncation, with advection in grid-point space based on the semi-Lagrangian scheme of Lin and Rood (1996).

The aerosol module HAM2 (Stier et al., 2005; Zhang et al., 2012) uses a two-moment modal approach to resolve the particle size distribution, which is based on M7 (Vignati, 2004) with four soluble and three insoluble modes. The particles in each mode are an internal mixture of up to five aerosol components (sulfate, sea salt, black carbon, particulate organic matter and mineral dust).

As in many models, clouds are divided into large-scale stratiform clouds and convective clouds. The former use a two-moment microphysics scheme (Lohmann et al., 2007; Lohmann and Hoose, 2009) with prognostic variables for liquid water content, ice content, CDNC and ice crystal number concentration (ICNC). Cloud cover fraction is diagnosed based on relative humidity (Sundqvist et al., 1989). In standard ECHAM–HAM, convective clouds are parmeterised with a bulk mass-flux scheme (Tiedtke, 1989; Nordeng, 1994) – we use this for "control" simulations, but our main results are based on replacing this with the Convective Cloud Field Model (CCFM) described in Section 2.2.

In this study, we use version ECHAM6.1–HAM2.2–MOZ0.9 in ECHAM–HAM configuration (i.e. with the MOZ chemisty switched off) at T63L31 resolution. This corresponds to about $1.875°$, with 31 vertical levels and a model top at $10\,\mathrm{hPa}$. As in Kipling et al. (2017), on top of the "standard" configuration, we use the Abdul-Razzak and Ghan (2000) aerosol activation scheme for stratiform clouds, including a multi-bin distribution of updraught velocities following West et al. (2014).

## 2.2 The Convective Cloud Field Model (CCFM)

CCFM is a spectral convective parameterisation which aims to represent the large-scale effects of an ensemble of multiple convective cloud types within each GCM column. This is based on the framework of Arakawa and Schubert (1974), coupled with an entraining plume model for each type of cloud with embedded aerosol activation and cloud microphysics. The heterogeneous cloud types are forced by their grid-scale environment, and each has an impact on this shared environment via entrainment and detrainment; these impacts can in turn produce a feedback on other cloud types. The resulting interactions, characterised in terms of competition for convective available potential energy (CAPE), form a set of Lotka–Volterra type equations. If convective quasi-equilibrium is assumed, these equations can be solved to determine the number of clouds of each type in the ensemble.

The individual cloud types are represented by a steady-state entraining plume model following Simpson and Wiggert (1969) and Kreitzberg and Perkey (1976), with an entrainment coefficient inversely proportional to the cloud radius. Cloud microphysics within the plume are calculated according to the one-moment bulk scheme used in ECHAM5 (Lohmann and Roeckner, 1996; Zhang et al., 2005). Without implementing a full two-moment microphysics scheme, this has been extended with a simple treatment of the evolution of CDNC within the rising parcel as in Labbouz et al. (2018) – whereas in Kipling et al. (2017) CDNC was assumed to remain constant from activation at cloud base to cloud top. In the revised scheme, the CDNC derived

from cloud-base activation is updated as the parcel rises with accretion, autoconversion, freezing and dilution by entrainment acting to reduce CDNC. (The activation of additional droplets above cloud base is not currently included).

The set of possible cloud types in CCFM is specified according to a range of starting radii between $200\,\mathrm{m}$ and the depth of the planetary boundary layer (PBL); an ensemble of 10 different entraining plumes is run over this range, each of which will develop differently as it rises in terms of its radius, velocity, moisture content and microphysics etc.

Further details of CCFM can be found in Wagner and Graf (2010) and Kipling et al. (2017).

## 2.3 Aerosol coupling

There are a number of mechanisms by which, without CCFM, ECHAM–HAM already supports aerosol effects on climate: direct radiative effects by scattering and absorption; semi-direct effects as cloud (both large-scale and convective) adjusts to the modified thermal profile; the cloud albedo effect due to changes in CDNC in the large-scale cloud scheme; and effects on (large-scale) cloud microphysics due to changes in CDNC, e.g. enhanced liquid water path due to rain suppression.

When using the standard Tiedtke–Nordeng convection scheme, there is no explicit coupling of aerosols and convection; however both the direct radiative effects and those on the large-scale cloud scheme (Lohmann et al., 2007) may nevertheless invoke a convective feedback. As mentioned above, Lohmann (2008) introduced aerosol-aware two-moment microphysics experimentally into this convection scheme, subject to the limited information available on vertical velocity and variation between clouds. With CCFM, these latter points can be taken into account much more explicitly since cloud-base vertical velocity is estimated directly from the sub-cloud triggering model, and the variation across the cloud spectrum represented explicitly (Kipling et al., 2017).

Activation at cloud base for each CCFM cloud type is calculated using the parameterisation of Abdul-Razzak and Ghan (2000), as in the large-scale cloud scheme, but using the convective-scale updraught velocity to determine CDNC at cloud base. This gives rise to two distinct types of aerosol–convection effects:

**microphysics effects** due to changes in droplet number propagating through the convective cloud microphysics, including changes in autoconversion rates, glaciation and associated latent heat release etc. (Heikenfeld et al., 2019), which affect the development of a given cloud type as the parcel rises, in turn potentially changing the balance of different cloud types in the competition for CAPE.

**anvil effects** due to changes in the number (and thus size) of droplets and ice particles detrained from CCFM into the large-scale cloud scheme, implemented as in Lohmann (2008) so that if convective cloud detrains with a greater CDNC or ICNC than that of any pre-existing large-scale cloud, then the large-scale CDNC or ICNC will be increased to match (though never reduced). This does not directly affect the development of the convective plumes but will alter their radiative effects, potentially feeding back on subsequent convection via changes to the thermodynamic profile and circulation. (The parameterised updrafts themselves do not interact with the radiation scheme directly, only via the detrained condensate as is the practice in many other models.)

In addition to these effects arising from aerosol interacting directly with convective cloud, there are the semi-direct effects due to aerosol–radiation interactions altering the thermodynamic profile and hence the CAPE and convective inhibition (CIN) in the environment. Particularly in the context of heating due to absorbing aerosol, these effects have been shown to have a significant effect on convective behaviour (e.g. Ackerman et al., 2000; Yamaguchi et al., 2015) which may dominate over the microphysical pathway in certain regimes.

In both the CCFMall_* and CCFMµphy_* simulations, the modelled CDNC is used in the microphysical parameterisations of the CCFM cloud model, enabling the microphysical effects; in CCFMall_* it also controls the number of droplets detrained into the large-scale cloud scheme, enabling the anvil effects, while in CCFMµphy_* only the bulk condensate is detrained without any explicit CDNC or size distribution (as when Tiedtke–Nordeng is used in standard ECHAM–HAM). In CCFMfix_*, the activation calculation is bypassed altogether; a fixed CDNC of $100\,\mathrm{cm}^{-3}$ is assumed in all CCFM convective clouds, and bulk condensate is detrained.

CCFM is only able to represent a subset of the possible aerosol effects on convective microphysics through changes in rain formation rates (autoconversion and accretion), and hence the amount of cloud water available to freeze. However, this pathway appears to be key to aerosol effects on convective clouds (Heikenfeld et al., 2019) and the approach is conceptually able to represent the thermodynamic invigoration proposed by Rosenfeld et al. (2008), even though it does not account for the separation of processes into latent heat release from freezing followed by off-loading of the condensate mass by falling precipitation Grabowski and Morrison (2016), and it is important to recognise the limitations of the Lagrangian entraining plume for representing other proposed mechanisms. In particular, the Lagrangian perspective cannot easily describe the non-instantaneous fall of precipitation through the plume as it rises, and CCFM does not currently attempt to do so. This means that, for example, the Fan et al. (2013) mechanism cannot possibly be captured since it hinges on changes to the fall speed. Similarly, in the absence of finite fall speeds, a proper representation of the melting of falling ice is not possible.

## 3 Experimental set-up

### 3.1 Simulations

In order to quantify the role of each of these mechanisms, including rapid adjustments and feedbacks, we have run the model in a number of different aerosol-coupling configurations which differ in the inclusion or exclusion of one of these mechanisms, as shown in Table 1.

Each configuration is run with both present-day (PD, year 2000) and pre-industrial (PI, year 1850) climatological emissions of aerosols and precursors, following the AeroCom Phase II/ACCMIP recommendations (http://aerocom.met.no/emissions.html). Only aerosol emissions differ between the simulations in each pair: greenhouse gases, ozone, sea-surface temperatures (SST) etc. are fixed to present-day climatology in all cases. In the case of SST and sea-ice, these use a 1979–2008 climatology of the AMIP2 dataset (https://pcmdi.llnl.gov/mips/amip/amip2/).

The simulations have each been run for a period of 10 years (plus 15 months of spin-up). To illustrate the aerosol perturbation that may lead to any aerosol–convection interactions, Figure 1 shows the differences between the CCN concentrations at the

surface, large-scale cloud base and convective cloud base in the PD and PI simulations. The concentrations at convective

cloud base look very similar to those at the surface: particles are mixed very rapidly throughout the PBL, and if convective cloud triggers in the model it does so very close to the PBL top. (For large-scale cloud on the other hand, the cloud base can sometimes be much more decoupled from the PBL in the model.) These highlight strong increases in CCN concentrations in China and India, but only moderate increases in other regions of convective activity (e.g. over the Amazon basin).

## 3.2   Analysis

The set of model configurations, with aerosol coupling processes successively activated, allows the response of any given model output to each process to be quantified via the difference between two of these configurations as detailed in Table 2.

We analyse this information from three different perspectives. Firstly, we look at the precipitation fields averaged over the whole 10-year period to identify the processes which affect the climatological distribution of precipitation in the model.

Secondly, we construct histograms showing the joint distribution of the radii and tops of the individual convective plumes

represented in CCFM (as introduced in Kipling et al., 2017), in order to understand the morphological responses at the convective scale in several regions with different convective regimes. The regions considered are the Amazon basin (continental deep convection), India and China (monsoon-driven, and with the largest PD−PI CCN increase) and the Caribbean (Atlantic trade cumulus). These are all defined as rectangular latitude–longitude boxes for simplicity, except that the "India" region is limited to the land points in the box using the model's land–sea mask to avoid conflating two quite different convective regimes.

In order to distinguish meaningful signals indicating aerosol effects from the noise due to the limited number of years in the simulations, we have carried out statistical significance testing at the 95% level using a two-sided paired samples $t$-test, where the PD and PI values for each year are treated as independent paired samples from their underlying distributions. This is visualised in the figures either by stippling over the areas of significance (for maps and 2D histograms) or by shading the corresponding confidence interval (for line plots).

## 4   Results

### 4.1   Mean precipitation response

The 10-year mean precipitation response to anthropogenic aerosol in the simulations with both Tiedtke–Nordeng and CCFM convection is shown in Figure 2. In addition to the "all effects" panel which shows the PD−PI difference with all feedbacks active, this is broken down into the separate large-scale and CCFM responses via the additional simulations as described

above. (For Tiedtke–Nordeng, only large-scale response mechanisms exist.) These effects are of a similar magnitude and in many regions they can oppose one another such that the overall precipitation change is not clearly driven by one mechanism in particular. CCFM and the large-scale effects drive a decrease in precipitation in China, while there is little change in India due to compensation between large-scale and CCFM effects; partial compensation is also observed over the Amazon. Tiedtke–Nordeng, with only large-scale effects represented, shows some differences in the regional pattern of responses (e.g. in India

and Australia), but also similarities in areas like China where the CCFM effect on precipitation is relatively weak. In the zonal mean, the difference remains noisy, with the only overall feature being a very small decrease in precipitation in the tropics coming from the large-scale effects – and even this has little statistical significance.

Figure 3 shows the fraction of total precipitation which comes from the convective parameterisation when using CCFM (rather than from the large-scale cloud and precipitation scheme), and also the PD−PI change due to all effects and due to the 225 convective microphysics alone. With CCFM, a slightly larger fraction of the total precipitation is produced in the convective parameterisation compared to Tiedtke–Nordeng (not shown), although this split is largely arbitrary and often varies from one model configuration to another (relating more to the difference between resolved and unresolved scales than the difference between observable cloud types). In both cases the fraction of tropical precipitation described as convective by the model is significantly larger than that classified as such by radar observations (Labbouz et al., 2018). In addition to the above point about 230 scale separation, this may also in part be due to the absence of mesoscale organisation with embedded stratiform precipitation in parameterised convection, one of the features highlighted by the Tropical Ocean–Global Atmosphere Coupled Ocean–Atmosphere Response Experiment (TOGA COARE; Houze, 1997; Halverson et al., 1999). The main pattern visible in the response of the convective precipitation fraction to PD aerosol loading is a small but statistically significant decrease over the continental northern mid-latitudes, coming from a combination of large-scale and CCFM effects (though the latter do not show 235 as significant in isolation), while a smaller decrease in the southern mid-latitudes results from the large-scale effects alone. This becomes clearer when looked at in the zonal mean.

It is important however to appreciate that the constraints on total precipitation are stronger in an atmosphere-only model with fixed SST such as this than they would be in a coupled atmosphere–ocean model where the SST is able to vary in response to a perturbation, providing additional mechanisms for feedbacks including changes to global evaporation rates.

## 4.2 Regional cloud field response

The explicit sub-grid-scale cloud fields of CCFM allow us to look more closely at the effects on simulated cloud morphology in a way that is not possible with a bulk mass-flux scheme like Tiedtke–Nordeng. The left column of Figure 4 shows the joint distribution of the maximum radius and cloud-top pressure of the parameterised clouds in four different regions (Amazon, India, China, Caribbean) with four additional regions shown in the supplement (Congo, Indonesia, SE Atlantic, SE Pacific); The right 245 column shows the PD−PI change in these distributions. The different regimes are clear, with the Amazon (characterised by continental deep convection) and India and China (characterised by monsoon-driven convection) showing significant greatest contribution of broad deep updraughts (upper right of the plots), and China showing the greatest proportion of very deep clouds. The SE Atlantic and Pacific (marine stratocumulus, in supplement) are confined to narrow, shallow updraughts (lower-left) as we might expect; the Caribbean (shallow convection, with occasional transition to deeper clouds) lies somewhere in between.

The change in these distributions between PI and PD aerosol conditions does not show a consistent pattern, but varies considerably between regions and regimes. Although there is some noise, many coherent features in the responses do show statistical significance. In the Amazon, we see a shift from broad deep clouds to narrower and shallower ones, either due to reduced development of the clouds or the triggering of clouds from smaller initial parcels. As there is no direct mechanism by

which aerosol can retard the development of an individual cloud in CCFM, this is likely to be the result of either reduced CAPE in the environment, or reduced CIN allowing smaller clouds to trigger. India is similar, but with only a slight lowering of the deepest clouds. In China, the dominant effect is simply an overall reduction in the amount of convective cloud (with the deep clouds most affected and only a slight increase in the smallest and shallowest clouds), suggesting a reduction of the large-scale convective forcing in the region – perhaps associated with changes to the monsoon circulation (e.g. Guo et al., 2013; Li et al., 2016) – or increased atmospheric stability due to the semi-direct effect of absorbing aerosol. The Caribbean, on the other hand, exhibits both a deepening of the main shallow-cloud regime, and also an increase in the small amount of much deeper cloud, suggesting some form of convective invigoration.

In order to disentangle these different responses, we turn to look at the separate contributions of the different mechanisms. Figure 5 shows these regional cloud field responses broken down into large-scale and convective mechanisms as listed in Table 2. (A further breakdown into ACI, ARI and the convective microphysics and anvil effects, is shown in the supplement.)

In the three deep convective regions (Amazon, India and China), the total aerosol effect on the cloud field is clearly dominated by that which occurs when only the large-scale mechanisms are active, as shown in the LS row of Figure 5. The additional contribution from the microphysical effects within CCFM (and their feedbacks) tends to partially counteract the effect of the large-scale changes, but the overall aerosol response remains qualitatively similar to the large-scale response alone. In addition, even where the contribution from the CCFM effects approaches a similar magnitude to that from the large-scale effects, the former show relatively little statistical signifance. Compensation between large-scale and microphysical aerosol effects have been observed in other recent studies, for instance in CRM simulations of the impact of biomass-burning aerosols on convection by Hodzic and Duvel (2018).

In the shallow-to-deep transition environment of the Caribbean, however, the weaker large-scale forcing leaves room for further vertical development of the convective cloud, and the total effect appears to be driven by the convective microphysics (as shown in the CCFM row of Figure 5) which in this region is statistically significant. The response via large-scale mechanisms alone is quite small and indistinct in comparison, with almost no statistical significance. Coupled with the results in the other regions, this suggests that where aerosols cause a change in the large-scale convective forcing or atmospheric stability this is the dominant effect. While direct impacts of the aerosol on convective microphysics are present, on larger scales their effects are overpowered by such changes to the "first-order" parameters controlling the convective behaviour. In particular, the quasi-equilibrium hypothesis ensures that, by construction, CCFM and other Arakawa–Schubert-type schemes are constrained to balance the large-scale forcing by the nature of their closure (at least in the mean, although changes in the distributions of cloud type and precipitation rate are still possible). Furthermore, the limitations of an atmosphere-only model mean that modulation of the large-scale forcing via SST feedbacks are not possible.

The idea that aerosol effects on convective microphysics are easily obscured by changes to the large-scale forcing is consistent with idealised studies (e.g. Lee et al., 2008; Fan et al., 2009) and the review of Fan et al. (2016), but is here demonstrated in the context of a global model with a full range of feedbacks and its own limitations. The stronger impact via convective microphysics in more weakly-forced shallow convection regimes like the Caribbean is in line with the conclusions of Storer et al. (2010) in a more idealised context. In a strongly-forced deep convective environment, there may be sufficient energy

input that glaciation is already occurring and any aerosol modulation of the latent heat release will have little effect; while in a more weakly-forced shallow-to-deep transition regime, such changes in latent heat release with the convective cloud may be enough to "tip" shallow clouds into transition or vice versa.

The changes to the vertical profile of rain and ice production (and associated latent heat release) within the CCFM-parameterised convective clouds (Figures 6 and 7) show that the response in the Amazon and China is again dominated by changes to the large-scale forcing, while in the Caribbean both are dominated by the response of the convective microphysics. The delaying of rain production to higher levels seen in the Caribbean, and greater ice production above, is exactly the signature of convective invigoration as described by Rosenfeld et al. (2008), suggesting that this can be a regionally important effect within a full global model, but only in a limited range of conditions. India is an interesting case, in that the overall effect on precipitation production is dominated by a reduction from the convective microphysics (not the anvil effects, as confirmed by Figures S5 and S6 in the supplement), even though the impact on the spectrum of cloud-top pressure is largely that of the large-scale forcing. This can be understood in terms of aerosol-suppressed autoconversion reducing warm rain production; however unlike in the Caribbean much of the cloud in this region is already glaciating, and the effect of additional rain suppression on the cloud field dynamics is much smaller in this case.

## 4.3 Implications for radiative forcing

The direct and indirect responses of parameterised convective clouds to aerosol have the potential to contribute positively or negatively to the overall effective radiative forcing due to aerosol–cloud interactions (ERFaci). These cannot in general be captured by the bulk mass-flux parameterisations commonly used in global climate models, and thus CCFM provides a novel and potentially useful tool for investigating their role from a modelling perspective at the global scale.

With Tiedtke–Nordeng, there is only a single class of aerosol–cloud interactions represented in the large-scale cloud and precipitation; using CCFM this contribution (ERFaci_ls) is joined by that due to interactions between aerosol and convection scheme itself (ERFaci_cv) if these are activated. These two combine to produce the total ERFaci. The extra ERFaci_cv seen in CCFM when these effects are activated is small and of marginal statistical significance from 10 years of simulation (95% confidence interval $-0.15 \pm 0.17 \, \mathrm{W} \, \mathrm{m}^{-2}$).

## 5 Conclusions

By explicitly considering convective microphysics and the sub-grid-scale heterogeneity of convective cloud, the Convective Cloud Field Model (CCFM) allows a physically-based parameterisation of aerosol–convection interactions to be included in a global atmospheric model. This extends the more usual state of the art, where only aerosol interactions with large-scale liquid clouds are explicitly represented in global models.

Using 10-year ECHAM–HAM–CCFM simulations with each of the interaction mechanisms (de-)activated in turn, we have shown how the different processes and feedbacks typically interact to produce an overall response. The global mean precipitation response is not dominated by one process, but results from a combination of convective and large-scale microphysics,

and feedback from aerosol–radiation interactions. To a large extent, these tend to counter one another, as expected based on the energetic control of global mean precipitation especially in an atmosphere-only mode with fixed SST. Investigation of how aerosol affects the distribution of precipitation intensity in the model, even if the total remains fixed, may be worth further study, as would the future extension to a coupled atmosphere–ocean model.

The impacts on cloud field morphology are also a combination of large-scale and convective mechanisms, with considerable regional variation. In the deep convective regions, the overall response is dominated by the combination of large-scale radiative and cloud effects (including their feedbacks on circulation), with a smaller countering contribution from convective microphysics (which is often not statistically significant) via changes in the vertical profiles of process rates within the CCFM cloud model. In the Caribbean shallow convection region, however, the response of the convective parameterisation itself to
the aerosol dominates and is statistically significant, with rain suppression, enhanced glaciation and deeper clouds indicative of convective invigoration.

    These results are consistent with previous more idealised studies which have suggested that shallower regimes with weaker forcing may be more susceptible to aerosol-induced invigoration than strongly-forced deep convection, and that aerosol microphysical effects become apparent only when they are not overpowered by the greater effect of changes to the large-scale
forcing (e.g. Lee et al., 2008; Fan et al., 2009; Storer et al., 2010). These conclusions have in general been based on idealised or limited-area models; this study shows evidence that they also hold in the context of the global atmosphere with all its feedbacks. However, the assumption of convective quasi-equilibrium in CCFM and many other convective parameterisations implicitly requires the large-scale forcing to be the dominant control on convection; at least a relaxation of this assumption may be required to investigate any cases where aerosol might have a dominant effect *despite* changes to the large-scale forcing.
However, the results also show that, allowing for feedbacks on convective forcing, the traditional invigoration hypothesis does not apply globally. This has implications in particular for nested convection-resolving simulations in which the large-scale forcing remains fixed, suppressing these feedbacks which may be key to the total response of the system to increased aerosol.

    From an effective radiative forcing perspective, a small additional effective forcing is seen from the aerosol–convective interactions captured in CCFM, but with 10 years of data this is of marginal statistical significance.

CCFM currently only represents a subset of the possible aerosol–convection interactions, especially in the context of mixed-phase microphysics; however these interactions appear to be of particular importance for the overall aerosol efffect on convection (Heikenfeld et al., 2019). Revision of the cloud model beyond the limitations of the Lagrangian entraining plume approach (which cannot include a proper treatment of hydrometeor sedimentation and accretion by rain or falling ice) to allow the incorporation of more sophisticated microphysics may open the way to capturing additional ice- and mixed-phase aerosol
effects. We have demonstrated, however, that CCFM is already able to identify a (modelled) susceptibility of one convective regime in particular (trade cumulus as found in the Caribbean) to aerosol-induced invigoration that is not damped or obscured by larger-scale dynamics. It is our hope that this hypothesis can be tested further by a comparison with detailed observations and cloud-resolving modelling on domains sufficiently large to capture these larger-scale feedbacks.

*Author contributions.* Zak Kipling designed and conducted the experiments and data analysis, and wrote the bulk of the manuscript. Laurent Labbouz developed some of the additional model code used, in particular relating to microphysics, provided insight during the analysis and additional text for the manuscript. Philip Stier provided supervision and guidance from the original concept, through the experiments and analysis to the drafting of the manuscript.

*Competing interests.* The authors declare that they have no conflict of interest.

*Acknowledgements.* The research leading to these results has received funding from the European Research Council under the European Union's Seventh Framework Programme (FP7/2007–2013)/ERC Grant Agreement FP7–280025 (ACCLAIM), from the European Union's Seventh Framework Programme (FP7/2007–2013) Project BACCHUS uder Grant Agreement 603445, and under the European Union's Horizon 2020 research and innovation programme with Grant Agreement 724602 (RECAP). The ECHAM–HAMMOZ model is developed by a consortium composed of ETH Zürich, Max Planck Intitut für Meteorologie, Forschungzentrum Jülich, the University of Oxford and the Finnish Meteorological Institute, and managed by the Center for Climate Systems Modeling (C2SM) at ETH Zürich. The authors would like to acknowledge the use of the University of Oxford Advanced Research Computing (ARC) facility in carrying out this work (http://dx.doi.org/10.5281/zenodo.22558). This work used the ARCHER UK National Supercomputing Service (http://www.archer.ac.uk/) and the UK Research Data Facility (http://www.archer.ac.uk/documentation/rdf-guide). L. Labbouz also received support from the French Space Agency (CNES).

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

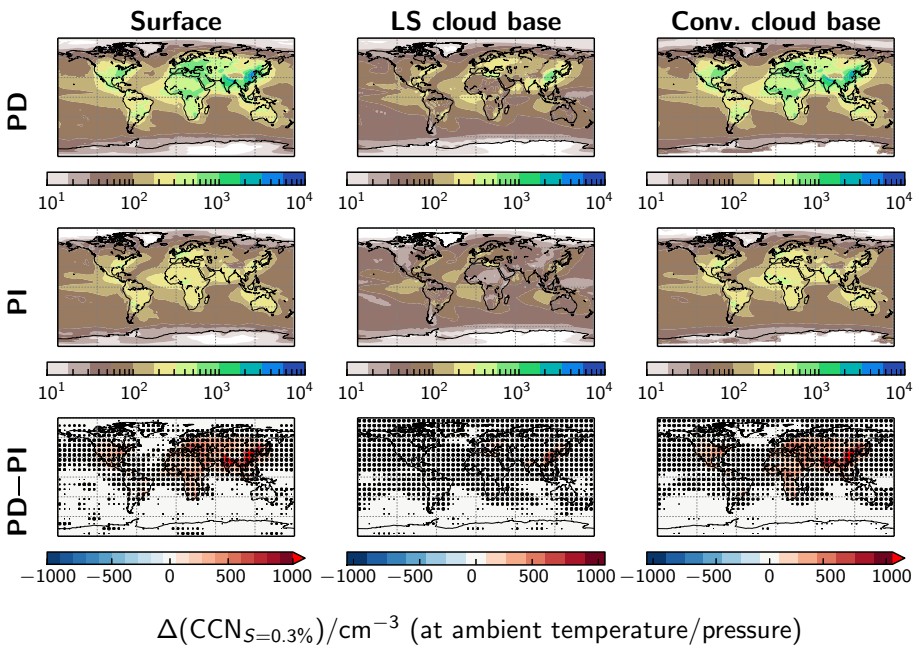

**Figure 1.** Annual mean CCN concentrations at the surface (left), large-scale cloud base (centre) and convective cloud base (PD−PI) from simulations using CCFM convection under PD (top row) and PI (middle row) aerosol emission scenarios, and the difference (bottom row). Specifically, these come from the CCFMall_ari simulations (see Table 1); however the difference in the others is small. (PD=present-day aerosol; PI=pre-industrial aerosol; stippling indicates areas where the PD−PI difference is statistically significant at the 95% level.)

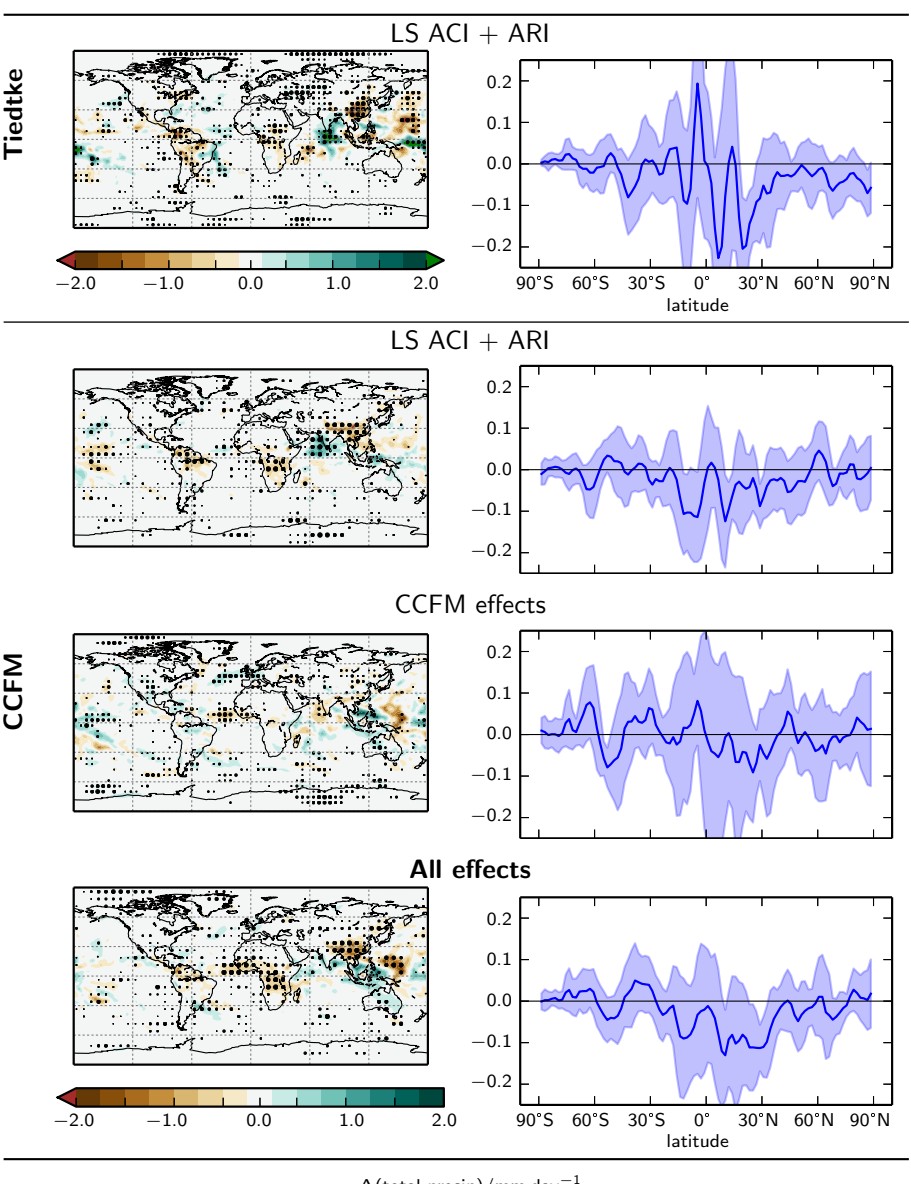

**Figure 2.** Annual mean precipitation response (PD−PI) in ECHAM–HAM with Tiedtke (top) and CCFM (below) convection, with the latter decomposed into large-scale and convective mechanisms as listed in Table 2. The plots on the right-hand side show the zonal mean of the maps on the left. A further breakdown into individual process effects is included in the supplement. (PD=present-day aerosol; PI=pre-industrial aerosol; stippling indicates areas where the PD−PI difference is statistically significant at the 95% level, while the corresponding confidence interval is shaded on the zonal mean plots.)

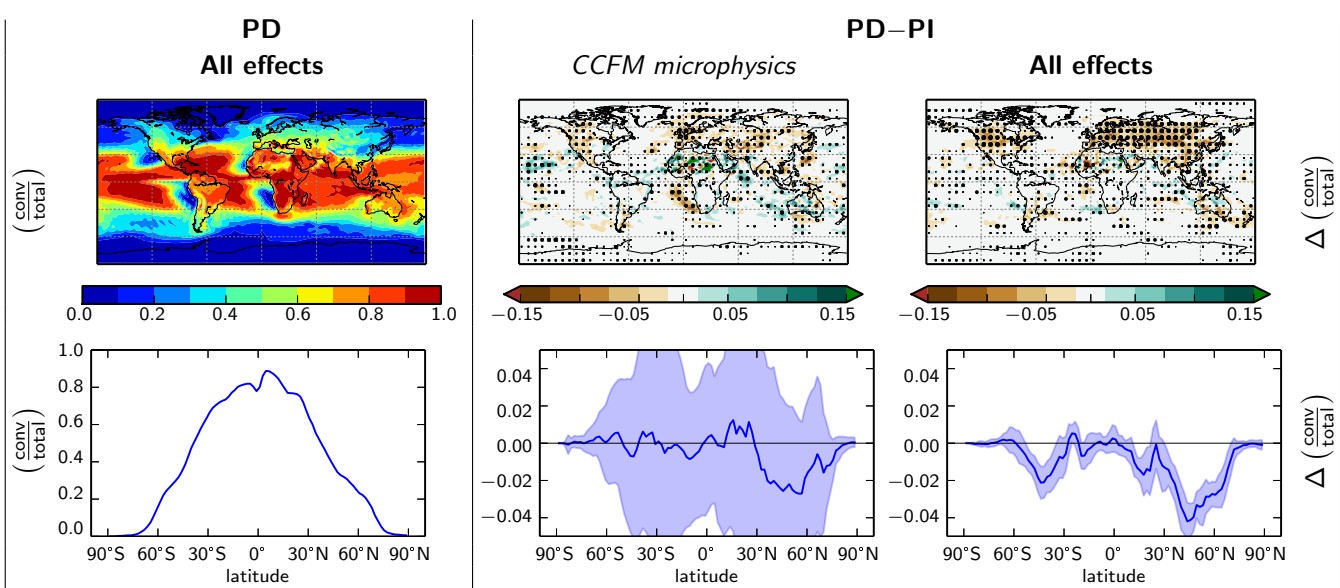

**Figure 3.** 10-year mean PD (left) and PD−PI change in (centre/right) fraction of precipitation which comes from CCFM (rather than resolved large-scale/stratiform cloud). The right-hand plot shows the total response, while the middle plot shows the component due to CCFM microphysics alone. The plots on the bottom row show the zonal mean of the maps above. (PD=present-day aerosol; PI=pre-industrial aerosol; stippling indicates areas where the PD−PI difference is statistically significant at the 95% level, while the corresponding confidence interval is shaded on the zonal mean plots.)

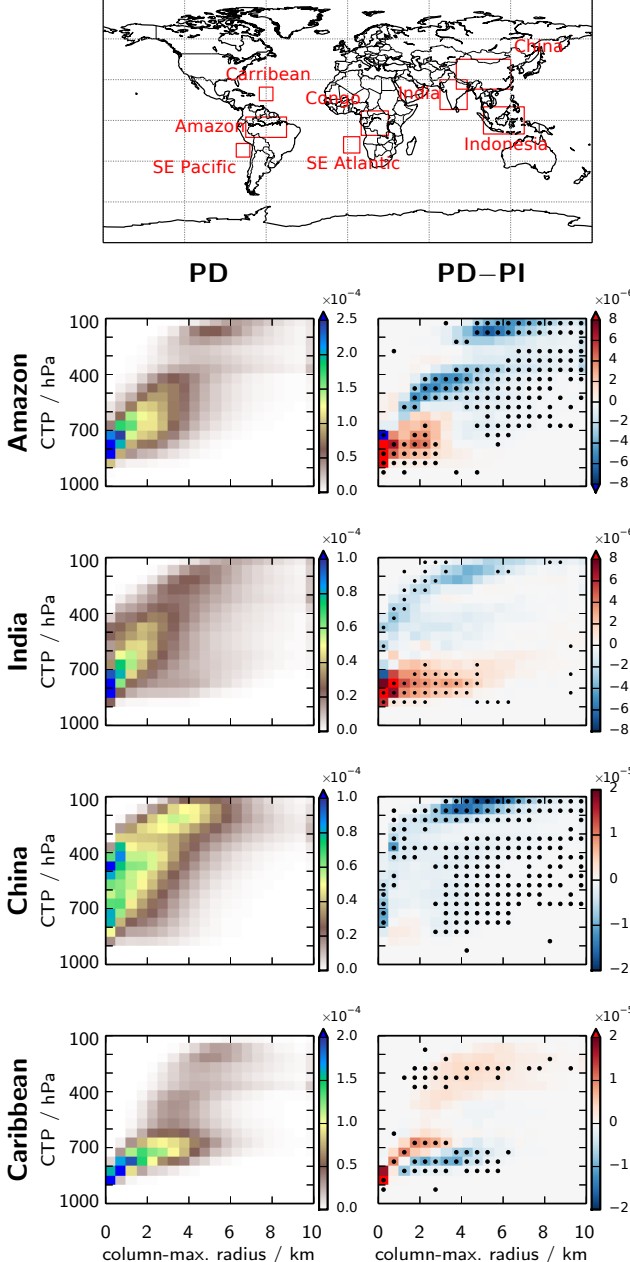

**Figure 4.** Response of the CCFM cloud top–radius distribution in four different regions, as shown in the map (top, along with four additional regions included in the supplement). Note that the "India" region used for analysis is restricted to the land points in the box. The left column shows the distribution under PD aerosol emissions; the right column shows the difference from PI aerosol emissions (with all effects included). The radius on the $x$-axis indicates the broadest part of a given entraining plume in the CCFM 10-member cloud-type ensemble over its whole height. (PD=present-day aerosol; PI=pre-industrial aerosol; black dots indicate histogram bins where the PD−PI difference is statistically significant at the 95% level.)

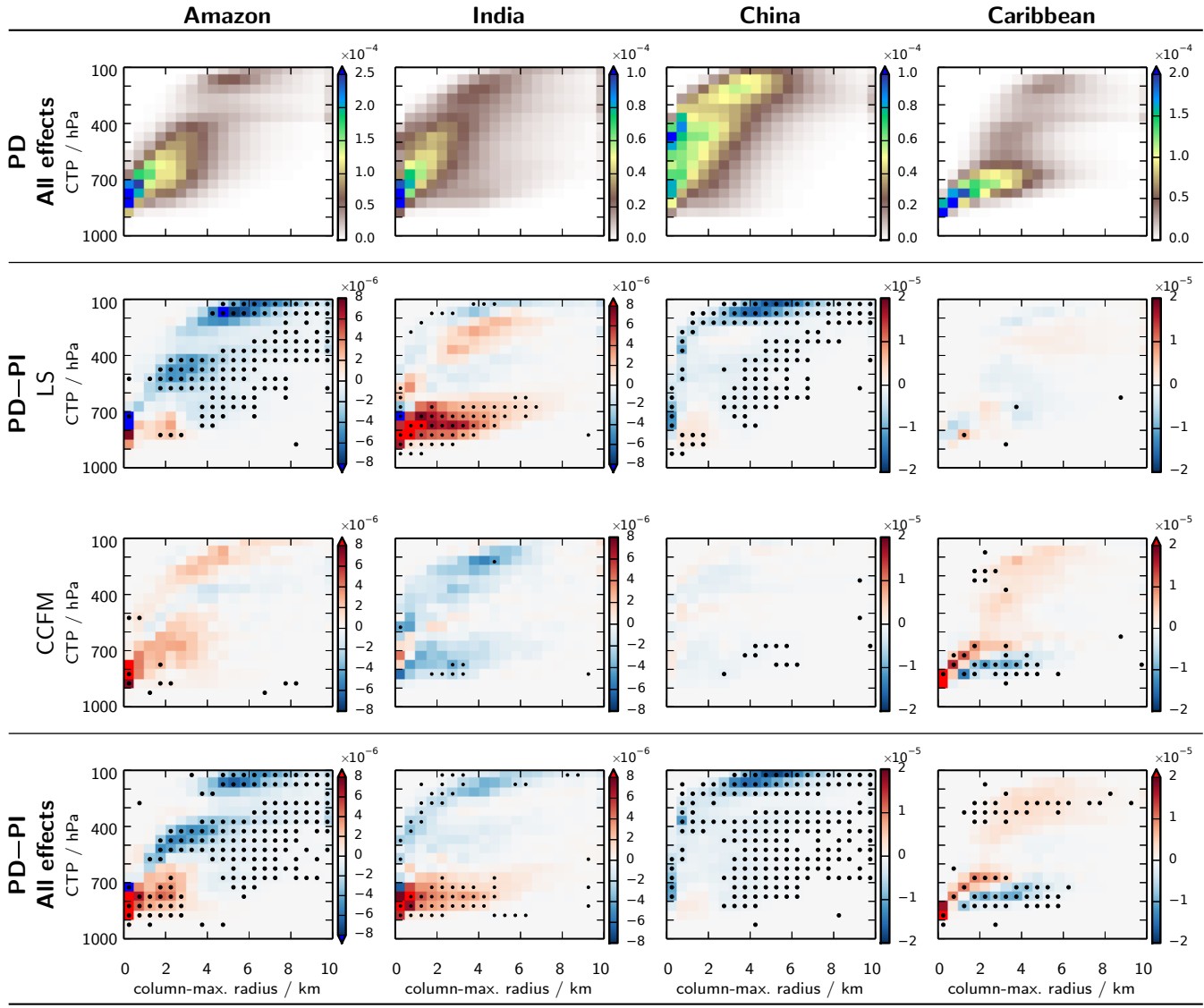

**Figure 5.** Regional cloud field response decomposed into its different mechanisms as listed in Table 2. The top row shows the PD distributions, while the bottom row shows the PD−PI difference (both with all effects included as in Figure 4); the middle rows show the contributions from large-scale and CCFM (convective) mechanisms. Note that the colour scales are identical between mechanisms to allow comparison of their magnitude, but not between the different regions. A further breakdown into individual process effects, along with additional regions, is included in the supplement. The radius on the $x$-axis indicates the broadest part of a given entraining plume in the CCFM 10-member cloud-type ensemble over its whole height. (PD=present-day aerosol; PI=pre-industrial aerosol; black dots indicate histogram bins where the PD−PI difference is statistically significant at the 95% level.)

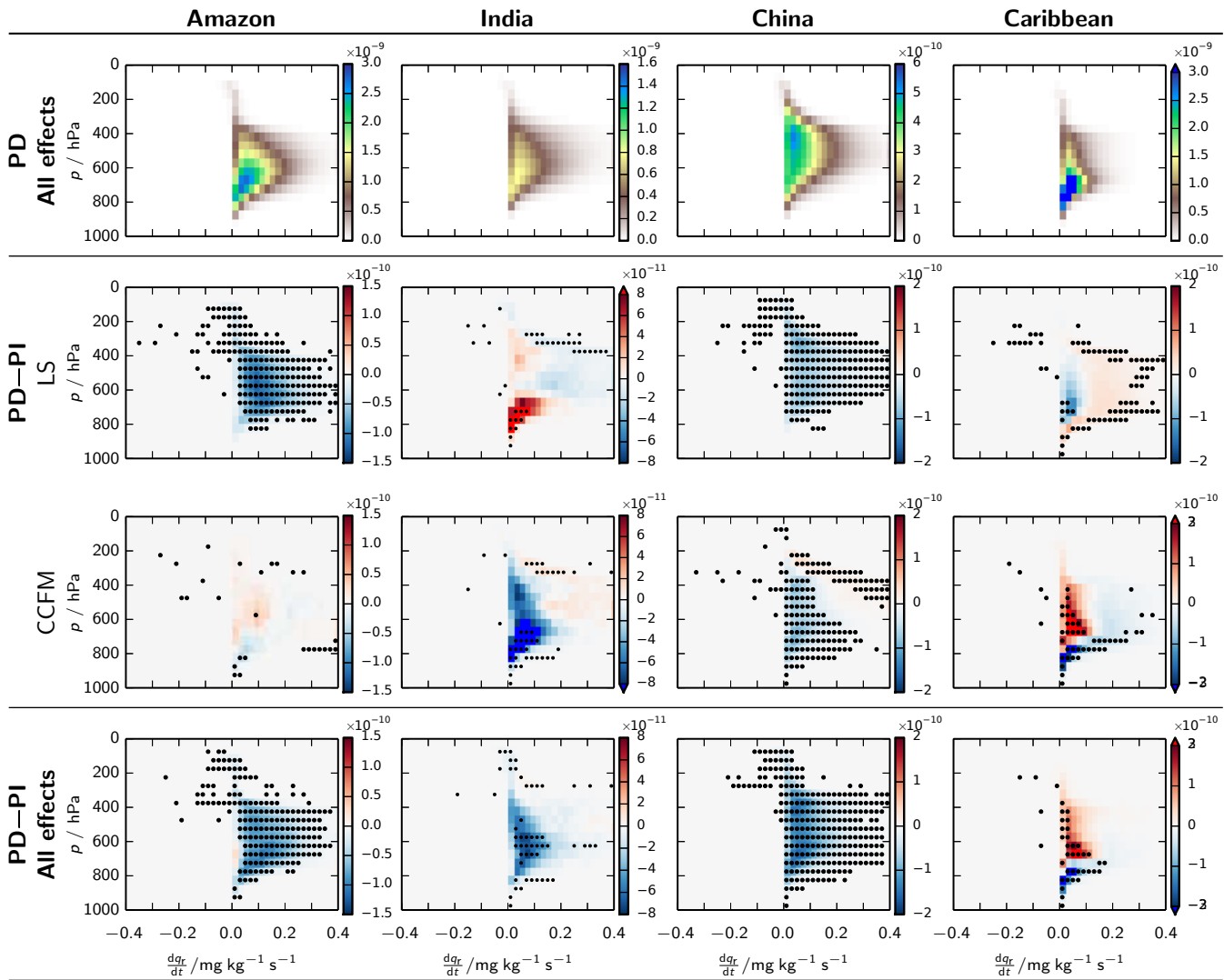

**Figure 6.** Regional change in the vertical profile of rain production within CCFM clouds, weighted by mass. The top row shows the PD distributions, while the bottom row shows the PD−PI difference (both with all effects included); the middle rows show the contributions from large-scale and CCFM (convective) mechanism. Note that the colour scales are identical between mechanisms to allow comparison of their magnitude, but not between the different regions. A further breakdown into individual process effects, along with additional regions, is included in the supplement. (PD=present-day aerosol; PI=pre-industrial aerosol; black dots indicate histogram bins where the PD−PI difference is statistically significant at the 95% level.)

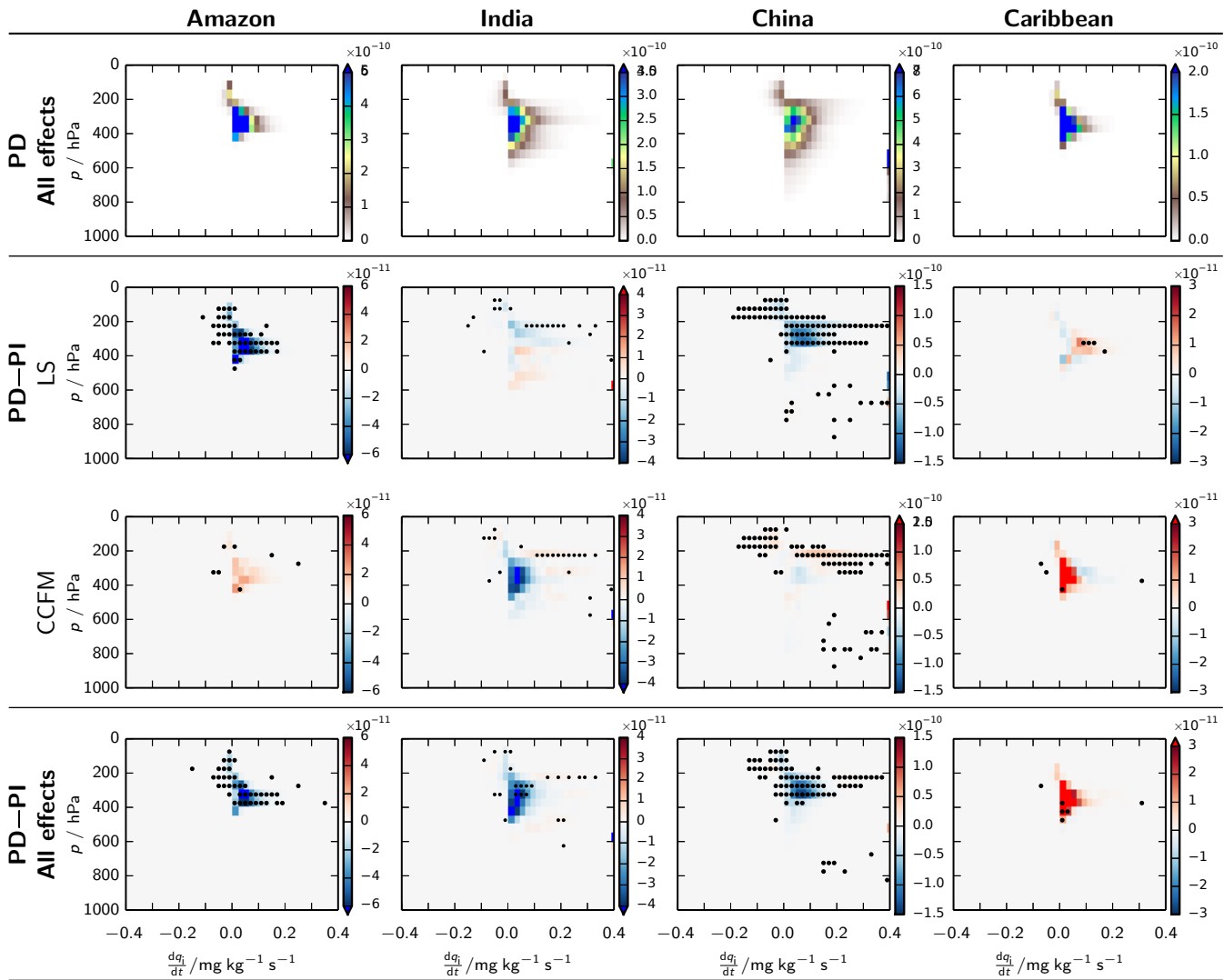

**Figure 7.** Regional change in the vertical profile of ice production (and consequent latent heat release) within CCFM clouds, weighted by mass. The top row shows the PD distributions, while the bottom row shows the PD−PI difference (both with all effects included); the middle rows show the contributions from large-scale and CCFM (convective) mechanism. Note that the colour scales are identical between mechanisms to allow comparison of their magnitude, but not between the different regions. A further breakdown into individual process effects, along with additional regions, is included in the supplement. (PD=present-day aerosol; PI=pre-industrial aerosol; black dots indicate histogram bins where the PD−PI difference is statistically significant at the 95% level.)

**Table 1.** Configurations used for aerosol effects in the ECHAM–HAM simulations. "Convective microphysics effects" refers to changes in microphysical process rates (autoconversion, accretion, freezing etc.) within the convective cloud as aerosol effects on droplet number propagate through the convective microphysics. "Convective anvil effects" refers to changes in the size distribution of droplets and/or ice particles detrained to the large-scale cloud scheme. (See Section 2.3 for further detail.)

| Label | Includes |
|---|---|
| **Tiedtke_noari** | large-scale cloud coupling only |
| **Tiedtke_ari** | as above + direct radiative effects |
| **CCFMfix_noari** | large-scale cloud coupling only |
| **CCFMfix_ari** | as above + direct radiative effects |
| **CCFMμphy_ari** | as above + convective microphysics effects |
| **CCFMall_ari** | as above + convective anvil effects |

**Table 2.** How the separate contributions of each mechanism to the total aerosol effect is extracted by taking the difference between pairs of simulations. (**ARI**: aerosol–radiation interactions; **LS ACI**: large-scale aerosol–cloud interactions; **CCFM microphysics**: changes to autoconversion etc. in convective cloud; **CCFM anvil**: changes to size distribution of detrained condensate. See Section 2.3 for more detail.

| Tiedtke | | |
|---|---|---|
| | LS ACI | = Tiedtke_noari |
| | ARI | = Tiedtke_ari − Tiedtke_noari |
| | **All effects** | = Tiedtke_ari |

| CCFM | | |
|---|---|---|
| | *LS ACI* | = CCFMfix_noari |
| | *ARI* | = CCFMfix_ari − CCFMfix_noari |
| | LS ACI + ARI | = CCFMfix_ari |
| | *CCFM microphysics* | = CCFMµphy_ari − CCFMfix_ari |
| | *CCFM anvil* | = CCFMall_ari − CCFMµphy_ari |
| | CCFM effects | = CCFMall_ari − CCFMfix_ari |
| | **All effects** | = CCFMall_ari |