# Peer review of "Global response of parameterised convective cloud fields to anthropogenic aerosol forcing"

_Atmospheric Chemistry and Physics, 2019_

## Referee Comment (RC1) · Anonymous Referee #1 · 27 Sep 2019

This paper addresses a neglected topic, the global impact of aerosols on the full range of convective clouds. While many climate models consider aerosol effects on clouds, they mostly consider this only on clouds formed by large-scale condensation in the model, and generally neglect the impact on convective (for example mixed-phase) clouds. This could be important not only by contributing to cloud radiative effects, but also by altering rainfall. Thus the current contribution is useful. The results do rely heavily on a convection scheme that may not represent the convective-scale dynamics very well, but the authors explain how this scheme works and how its assumptions play into the results obtained; the study is a useful advance even though conditional on the scheme. The results qualitatively support previous suggestions based on more

localised modelling approaches.

My chief concern with this paper is that I am not sure 10 years is long enough to see the impact of aerosols on rainfall, due to its high internal variability. This impact is subtle compared to the large dynamical influences on convection, as acknowledged by the authors, and the results are not straightforward, with opposite-signed changes in different regions that defy any simple explanation as an aerosol impact and look noisy. I don't see any significance testing applied to the claimed impacts, but the authors need to do this– for example by considering each year as an independent sample and computing a t-statistic on the differences in mean rainfall or cloud property, either at the pixel or region level or aggregated in some other way to improve the statistics.

Detailed comments.

86: Please clarify whether new particle formation is included in this model. Here it says the gas-phase chemistry model is not active for this study, but later (l174) the manuscript speaks of precursor emissions, which would seem to be irrelevant if there is no gas-phase chemistry or new particle formation predicted.

175-ish: please specify what SSTs were used (are they the same each year for the 10 years?)

212-214: It is a common misconception that precipitation generated by the convective scheme is "convective" and that generated by the grid-scale scheme is "stratiform." The convective scheme simply represents the net effect of all air motions below the grid scale. For example as the model resolution increases, the fraction of total P produced by the convective scheme should steadily decrease, but the model is not predicting that rain is physically becoming more stratiform. Since convective motions in nature mathematically project onto the grid scale circulation, some of the grid-scale condensation will be attributable to convection, no matter what the resolution.

Fig 1. caption: might be helpful to (re)define PD and PI in the caption.

[Figure]

---

## Referee Comment (RC2) · Anonymous Referee #2 · 23 Oct 2019

Kipling et al. use a model with parameterized deep convection to study aerosol effects on deep convection. They do so in an atmosphere-only model setup that does not allow sea surface temperatures (SSTs) to respond to the aerosol radiative forcing. The main advantage of this study is that they take into account feedbacks from the large scale circulation (or at least those effects which are not mediated by SST changes). Unfortunately, however, in spite of many nice and useful diagnostics, the discussion of aerosol effects in the strongly forced regions ultimately still seems fairly superficial. I think that in order to better understand the overall effect of taking into account aerosol-deep-convection interactions, potential reasons for the qualitative difference of the CCFM effects between the India region and the Amazon region should be explored

in this study. A specific suggestion can be found below, but I hope that perhaps the authors can add to this. To me, the explanation that "[i]n a strongly-forced deep convective environment, there may be sufficient energy input that any aerosol modulation of the latent heat release will have little effect" seems overly simplistic regarding the qualitative difference between the CCFM effects in the strongly forced regions in Figure 5. I think that after some additional analysis to address the differences between aerosol effects in strongly forced regions, this study will ultimately be a useful addition to the literature on the interesting and important topic of aerosol effects on deep-convection. I recommend major revisions. It should be noted that convective invigoration could in principle affect precipitation intensity while the time mean precipitation amount remains constant.

Major comments:

1) India region: I suggest to analyze ocean and land in the India region separately. As far as I can see, the change regarding LS ACI + ARI in the India region in the Figure S5 is at odds with many studies of aerosol effects over India. Figure 2 suggests that this could in part be due to changes in the Arabian Sea and the Bay of Bengal.

2) Amazon region: As far as I can see, Figure S2 suggests some invigoration in CCFM-microphys in the Amazon region which is partially compensated by CCFM-anvil. Please discuss.

3) I think that the reasons for the apparent convective suppression due to including aerosol effects in CCFM in the India region should be analyzed and discussed in more detail.

4) Because the CCFM effects are defined as CCFMall_ari-CCFMfix_noari, I don't understand why the CCFM effects over India still seem to compensate the large scale effects in such a rather systematic manner (see Fig. 2 and Fig. 6, discussion in line 294 and elsewhere). Please explain.

Minor comments:

l. 6: what does "explicitly simulate" mean here?

l. 12: "rather small" compared to large-scale effects? Fig.6 suggests similar magnitudes for India.

l 25: Yes, "total precipitation is constrained to balance evaporation and by energetic constraints". The problem here is that sea surface temperatures are fixed. In coupled climate models aerosol does affect sea surface temperature and consequently also evaporation. When the sea surface temperatures cannot respond to the aerosol forcing, a large part of the precipitation response that one would see in a coupled climate model is lost. In other words, it is not possible to fully simulate the effects of aerosol on global mean precipitation in the setup used here. I think this point should already be stressed in the introduction and later also be taken into account in the results section (especially in Sect. 4.1) as well as in the conclusion section.

l 310: it seems odd to me that the effect of aerosol on global mean precipitation is emphasized in the conclusion section. When SSTs are allowed to respond to forcing, global mean precipitation changes. When it is not, global mean precipitation stays more or less constant. An exception to this are model runs in which the forcing is from greenhouse gases which by definition absorb longwave radiation. Greenhouse gases affect precipitation more directly than scattering aerosol via their influence on the radiation budget. I think that instead of discussing changes of global mean precipitation in quite some detail, the consequences of keeping SSTs fixed should be discussed. It should also be noted somewhere that convective invigoration could affect precipitation intensity while time mean precipitation remains constant.

l. 256: quasi-equilibrium: I am not convinced that this is actually a problem here since the deep convective heating is applied to the large scale circulation and since large scale water vapor is also modified. More active deep convection may result in less large-scale condensation, but there would still be a shift in precipitation production

from "stratiform" cloud microphysics to the deep convection parameterization, which would then constitute the signature of deep convective invigoration. In my opinion, fixed SSTs are potentially more problematic than using the quasi-equilibrium assumption in the deep convection parameterization.

l. 1: references?

l. 33: "we are probably decades away from having the computer power to run convection-resolving global climate models for more than short time periods": Could you please cite a reference or provide some backing for this statement in the reply to this comment. This does not have to be included in the manuscript.

l. 67: "still lacking non-local interactions": I reviewed Wang et al. (2011), but I don't understand what the authors are referring to. Please explain.

l. 109: what does "explicit" mean here? Maybe explain briefly how this differs from A+S74.

l. 142: if I understand this right, the anvil effects includes the effect from detrainment into the large scale. Does this have consequences for the separation between CCFM and large scale effects? Please clarify.

l. 142: please also explain briefly how the effect of detrainment on CDNC (and ICNC?) is treated.

l. 156: a fixed CDNC is assumed in all CCFM convective clouds <- how many CDNC?

l. 214f: based on my understanding, having a more inactive deep convection scheme leads to more stratiform precipitation and vice versa. Sometimes, this behavior can be influenced by tuning parameters in the convection scheme. The potential explanation given here is, however, fine with me.

l. 263f: I don't understand how this explanation fits with the CCFM results for India in Figure 5 and 7.

l. 301: Please repeat which idealized studies this refers to.

l. 301: I can see a number of potential problems with limited area studies, and I think that it is important to perform the type of study presented here. For example, applying a large scale forcing in a limited-area model means that in a strongly forced case, precipitation will more or less simply equal the large scale vapor large scale forcing. At the same time, it is not directly clear to me why phase changes due to precipitation formation delays should be less important in a strongly forced case, unless CCN are depleted by precipitation.

l. 309: what does "via energetic and water-budget constraints" mean? Please be more specific.

l. 310: Although invigoration might not change the time average amount of precipitation, it could still have an impact on precipitation intensity. Please discuss.

Fig 1: does this depend strongly on the model run?

Fig. 4: I think it would be good to explain the x-axis better. Perhaps you could mention the number of ensemble members in CCFM somewhere. I would like to obtain a rough idea about how the values on the x-axis come about without having to consult additional literature.

Technical:

l. 98: uses -> is diagnosed (and remove diagnostic in line 99)

l. 177: months' -> should it just be months without the apostrophe?

---

## Author Comment (AC1) · 2 Mar 2020

**Response to Referee Comments on "Global response of parameterised convective cloud fields to anthropogenic aerosol forcing"**

Z. Kipling, P. Stier, and L. Labbouz

March 2, 2020

We are grateful to the two anonymous referees for their time and constructive comments on our original manuscript and during the public discussion. We have made a number of alterations in a revised manuscript to address the points raised by both reviewers during the discussion phase, and we hope that the manuscript is now clearer as a result. Responses to individual points, and details of changes to the manuscript, are given below.

**Response to reviewer #1**

*This paper addresses a neglected topic, the global impact of aerosols on the full range of convective clouds. While many climate models consider aerosol effects on clouds, they mostly consider this only on clouds formed by large-scale condensation in the model, and generally neglect the impact on convective (for example mixed-phase) clouds. This could be important not only by contributing to cloud radiative effects, but also by altering rainfall. Thus the current contribution is useful. The results do rely heavily on a convection scheme that may not represent the convective-scale dynamics very well, but the authors explain how this scheme works and how its assumptions play into the results obtained; the study is a useful advance even though conditional on the scheme. The results qualitatively support previous suggestions based on more localised modelling approaches.*

*My chief concern with this paper is that I am not sure 10 years is long enough to see the impact of aerosols on rainfall, due to its high internal variability. This impact is subtle compared to the large dynamical influences on convection, as acknowledged by the authors, and the results are not straightforward, with opposite-signed changes in different regions that defy any simple explanation as an aerosol impact and look noisy. I don't see any significance testing applied to the claimed impacts, but the authors need to do this – for example by considering each year as an independent sample and computing a t-statistic on the differences in mean rainfall or cloud property, either at the pixel or region level or aggregated in some other way to improve the statistics.*

This is an excellent point, and for all the plots illustrating PD−PI differences we have now included an indication of statistical significance at the 95% level, based on a two-sided $t$-test. The value for each of the 10 years is treated as an independent sample from the underlying distribution, as suggested by the reviewer, with the PD and PI values for a given year being matched in a "paired-samples" t-test.

For maps of differences, the areas of significance are shown with stippling; likewise for the 2D histograms a dot is placed in each shaded bin where the difference is significant. For the zonal-mean difference plots, the corresponding 95% confidence interval is shaded, indicating significance where the shaded region does not cross zero.

This analysis reassuringly shows that many (though not all) of the coherent features that show up in the results do indeed have statistical significance based on 10 years of simulation. We have updated the text to describe the methodology applied (extending Section 3.2), and to indicate the statistical significance of the major features in the results to which the discussion refers (throughout Sections 4.1 and 4.2), and metion this in the conclusions.

We also believe that identifying the statistically significant responses, and separating them from the noise, helps to address some of the points raised by Reviewer #2 (see below).

*Detailed comments.*

*86: Please clarify whether new particle formation is included in this model. Here it says the gas-phase chemistry model is not active for this study, but later (l174) the manuscript speaks of precursor emissions, which would seem to be irrelevant if there is no gas-phase chemistry or new particle formation predicted.*

The full MOZ chemistry scheme is not included in the model configuration used here; however the ECHAM–HAM configuration still contains a simple sulphate precursor chemistry driven by DMS and $SO_2$ emissions. Oxidation from DMS to $SO_2$ to $H_2SO_4$ follows a simplified parameterisation in the absence of full oxidant chemistry, and sulphate aerosol can then be produced by new particle formation, condensation of $H_2SO_4$ onto existing particles, and by parameterised aqueous oxidation within cloud droplets. We have clarified this point in the text.

*175-ish: please specify what SSTs were used (are they the same each year for the 10 years?)*

Sea surface temperatures use a monthly climatology derived from years 1979–2008 of the AMIP2 SST dataset, so the interannual variability is not included here. We have clarified this in the text.

*212–214: It is a common misconception that precipitation generated by the convective scheme is "convective" and that generated by the grid-scale scheme is "stratiform." The convective scheme simply represents the net effect of all air motions below the grid scale. For example as the model resolution increases, the fraction of total P produced by the convective scheme should steadily decrease, but the model is not predicting that rain is physically becoming more stratiform. Since convective motions in nature mathematically project onto the grid scale circulation, some of the grid-scale condensation will be attributable to convection, no matter what the resolution.*

The reviewer is of course correct, in that the difference between "convective" and "stratiform" in a model such as this is really one of resolved vs sub-grid-scale motions, and does not necessarily correspond to physical cloud types. It is not our intention to suggest otherwise, and this is why we make clear that we are talking about the split between the *paramteterisations*, and that this is somewhat arbitrary; however we have updated the text to make this point more explicit.

*Fig 1. caption: might be helpful to (re)define PD and PI in the caption.*

We have done this as suggested.

**Response to reviewer #2**

*Kipling et al. use a model with parameterized deep convection to study aerosol effects on deep convection. They do so in an atmosphere-only model setup that does not allow sea surface temperatures (SSTs) to respond to the aerosol radiative forcing. The main advantage of this study is that they take into account feedbacks from the large scale circulation (or at least those effects which are not mediated by SST changes). Unfortunately, however, in spite of many nice and useful diagnostics, the discussion of aerosol effects in the strongly forced regions ultimately still seems fairly superficial. I think that in order to better understand the overall effect of taking into account aerosol-deep-convection interactions, potential reasons for the qualitative difference of the CCFM effects between the India region and the Amazon region should be explored in this study. A specific suggestion can be found below, but I hope that perhaps the authors can add to this. To me, the explanation that "[i]n a strongly-forced deep convective environment, there may be sufficient energy input that any aerosol modulation of the latent heat release will have little effect" seems overly simplistic regarding the qualitative difference between the CCFM effects in the strongly forced regions in Figure 5. I think that after some additional analysis to address the differences between aerosol effects in strongly forced regions, this study will ultimately be a useful addition to the literature on the interesting and important topic of aerosol effects on deep-convection. I recommend major revisions. It should be noted that convective invigoration could in principle affect precipitation intensity while the time mean precipitation amount remains constant.*

*Major comments:*

*1) India region: I suggest to analyze ocean and land in the India region separately. As far as I can see, the change regarding LS ACI + ARI in the India region in the Figure S5 is at odds with many studies of aerosol effects over India. Figure 2 suggests that this could in part be due to changes in the Arabian Sea and the Bay of Bengal.*

The reviewer makes a good point that the change in this region could be a mixture of quite different effects over the land and ocean portions of the box. To answer this, we have therefore repeated the analysis restricted to "land" points using the model's land–sea mask (see Figures A1, A2 and A3 in this response. This does indeed appear to remove an additional set of large-scale processes that are showing up in the cloud-field and rain-production responses. Because this makes the overall results cleaner, we have changed the definition of the "India" region to refer to the land only (which was always the intended focus rather than the ocean which falls into the same rectangular box).

*2) Amazon region: As far as I can see, Figure S2 suggests some invigoration in CCFM-microphys in the Amazon region which is partially compensated by CCFM-anvil. Please discuss.*

The changes in the Amazon region due to CCFM show only marginal statistical significance, and look quite noisy/patchy over the CTP-radius distribution rather than indicating any clear invigoration or suppression effect in our view. The large-scale effects, on the other hand, show a quite coherent suppression of deep convection overall, consistent with our interpretation that aerosol effects on convection in the model in this region are being driven primarily by modulation of the large-scale forcing. In updating the text to refer to significance following Reviewer #1's comments, we have noted in the text that the effects in the deep convective regimes are mostly noisy and not statistically significant.

*3) I think that the reasons for the apparent convective suppression due to including aerosol effects in CCFM in the India region should be analyzed and discussed in more detail.*

Again, this is a case where there is little statistical significance to the apparent signal coming from CCFM effects in Figure 5; the only regions where a coherent significant effect is visible are the Caribbean (a clear invigoration signal that dominates the total effect), and the Congo (a smaller but significant invigoration that is nevertheless outweighed by the large-scale effects).

Although the cloud field morphology changes in Figure 5 are not that significant here, there *is* a clear decrease in rain production from the CCFM microphysics effects in Figures 6 and S5. This can be explained in terms of aerosol increasing CDNC and suppressing autoconversion, although it is interesting that the effect on the cloud field is weak here (unlike in the Caribbean case where suppression of autoconversion leads to clear convective invigoration). However, this fits with the idea that aerosol can push shallower cloud into transition, but its effect on already-deep cloud regimes is more limited overall even if it does reduce warm rain production.

We have updated the final paragraph of Section 4.2 to take account of this.

*4) Because the CCFM effects are defined as CCFMall_ari−CCFMfix_noari, I don't understand why the CCFM effects over India still seem to compensate the large scale effects in such a rather systematic manner (see Fig. 2 and Fig. 6, discussion in line 294 and elsewhere). Please explain.*

Note that the CCFM effects are defined as CCFMall_ari−CCFMfix_ari, as per Table 2 (not CCFMfix_noari as stated in the comment).

This is not specifically a feature of the behaviour over India, but the same partial compensation which is noted in many of the regions, with one scale of processes tending to oppose the other, although it is the large-scale processes which dominate the overall response in most cases.

*Minor comments:*

*l. 6: what does "explicitly simulate" mean here?*

We have changed this to "simulate…complete with microphysics and interactions between clouds" for clarity.

*l. 12: "rather small" compared to large-scale effects? Fig.6 suggests similar magnitudes for India.*

This statement is talking about the convective cloud field morphology, as shown in Figure 5, which is affected much more strongly by the large-scale effects than those via CCFM itself. The reviewer is correct, however, that there is nevertheless a strong decrease in rain production (Figure 6) from the CCFM effects that does not translate into a significant modification of the cloud field itself, which we have discussed above.

*l 25: Yes, "total precipitation is constrained to balance evaporation and by energetic constraints". The problem here is that sea surface temperatures are fixed. In coupled climate models aerosol does affect sea surface temperature and consequently also evaporation. When the sea surface temperatures cannot respond to the aerosol forcing, a large part of the precipitation response that one would see in a coupled climate model is lost. In other words, it is not possible to fully simulate the effects of aerosol on global mean precipitation in the setup used here. I think this point should already be stressed in the introduction and later also be taken into account in the results section (especially in Sect. 4.1) as well as in the conclusion section.*

We fully agree that there are limitations in the feedback processes which can be captured in an uncoupled atmosphere-only model (especially with respect to those operating on longer timescales and for maritime convection). Nevertheless, the approach still allows many inter-scale feedbacks via atmospheric processes alone to be represented, and it is more the effects on *regional* precipitation and cloud fields than on global-mean precipitation which are the focus of this paper. We have updated the text to acknowledge the limitations of an atmosphere-only model in this context, and suggest that further studies using a coupled atmosphere–ocean model might provide further insight, though with a greater computational cost (final paragraph of introduction, extending Section 4.1, and in the second paragraph of the conclusions).

*l 310: it seems odd to me that the effect of aerosol on global mean precipitation is emphasized in the conclusion section. When SSTs are allowed to respond to forcing, global mean precipitation changes. When it is not, global mean precipitation stays more or less constant. An exception to this are model runs in which the forcing is from greenhouse gases which by definition absorb longwave radiation. Greenhouse gases affect precipitation more directly than scattering aerosol via their influence on the radiation budget. I think that instead of discussing changes of global mean precipitation in quite some detail, the consequences of keeping SSTs fixed should be discussed. It should also be noted somewhere that convective invigoration could affect precipitation intensity while time mean precipitation remains constant.*

It is not our intention to "emphasize" the effect on global mean precipitation; rather we present this as the global context in which the regional precipiation and cloud-field changes which *are* the focus of the manuscript are discussed. We hope that the changes made in respect of the previous point, acknowledging that the overall balancing-out of the global-mean response follows from the use of fixed SSTs, address this concern. The possibility to modulate precipitation intensity while keeping total precipitation fixed is already mentioned at the end of the first paragraph of the introduction.

*l. 256: quasi-equilibrium: I am not convinced that this is actually a problem here since the deep convective heating is applied to the large scale circulation and since large scale water vapor is also modified. More active deep convection may result in less large-scale condensation, but there would still be a shift in precipitation production from "stratiform" cloud microphysics to the deep convection parameterization, which would then constitute the signature of deep convective invigoration. In my opinion, fixed SSTs are potentially more problematic than using the quasi-equilibrium assumption in the deep convection parameterization.*

The problems of fixed SSTs and the quasi-equilibrium hypothesis are somewhat similar, in that both will limit the feedback mechanisms that are captured by the model (though of course in different ways). We have updated the text to reflect the fact that fixed SSTs may also be a limitation here.

*l. 1: references?*

Appropriate references are those cited in the first paragraph of the introduction – in the interest of conciseness, however, we do not believe it is appropriate to duplicate them in the abstract.

*l. 33: "we are probably decades away from having the computer power to run convection-resolving global climate models for more than short time periods": Could you please cite a reference or provide some backing for this statement in the reply to this comment. This does not have to be included in the manuscript.*

This is our impression of the current state of global climate modelling, however in the absence of a specific reference, we've revised this to "we are some way from having the computer power to routinely run convection-resolving global climate models for more than short time periods" which we hope is less controversial than estimating how long it might take to reach that point.

*l. 67: "still lacking non-local interactions": I reviewed Wang et al. (2011), but I don't understand what the authors are referring to. Please explain.*

In the superparameterisation approach, although convective elements are explicitly resolved on e.g. a 2D grid within each host-model column, they are able to interact only "locally", i.e. between elements within the same host-model column – interactions with convective elements in neighbouring columns occurs only via the host model fields. We have updated the text to explain our understanding of the non-locality issue.

*l. 109: what does "explicit" mean here? Maybe explain briefly how this differs from A+S74.*

We have changed this to "an entraining plume model for each type of cloud with embedded aerosol activation and cloud microphysics" for clarity.

*l. 142: if I understand this right, the anvil effects includes the effect from detrainment into the large scale. Does this have consequences for the separation between CCFM and large scale effects? Please clarify.*

Yes, that is exactly the effect that is being referred to here – changes to the droplet/ice-particle size at detrainment will not directly affect the convective clouds themselves, but will feed back by changing the microphysics in the large-scale cloud scheme, thus acting as a modulation of the LS ACI effects.

This highlights the fact that there isn't a clean separation between the effects – as each of them is "switched on" going down the list in Table 1, it may trigger feedbacks by modifying the processes above which are already active, as well as its own direct impact.

*l. 142: please also explain briefly how the effect of detrainment on CDNC (and ICNC?) is treated.*

The two-moment detrainment, explicitly passing a number of droplets or ice crystals to the large scale rather than just a bulk mass, is implemented using the same mechanism as in Lohmann (2008): if a convective cloud detrains liquid or ice with a number concentration greater than that of any existing large-scale cloud, the large scale CDNC or ICNC will be increased to match (though it will never be decreased by convective detrainment). We have added the explanation in the text.

*l. 156: a fixed CDNC is assumed in all CCFM convective clouds <- how many CDNC?*

The fixed value is $100 \, \text{cm}^{-3}$ globally in both PD and PI CCFMfix simulations. We have clarified this in the text.

*l. 214f: based on my understanding, having a more inactive deep convection scheme leads to more stratiform precipitation and vice versa. Sometimes, this behavior can be influenced by tuning parameters in the convection scheme. The potential explanation given here is, however, fine with me.*

Yes, given that (as discussed elsewhere) the total precipitation is constrained, any change in tuning parameters that alters the amount from one scheme must cause a balancing change in the other. The tuning of this balance is certainly something which could be the subject of further investigation, but is not a focus of the present study.

*l. 263f: I don't understand how this explanation fits with the CCFM results for India in Figure 5 and 7.*

As discussed above, the CCFM effect on the cloud field itself (Figure 5) is small compared to the large-scale effects. Figures 6 and 7 show that microphysical effects are changing the autoconversion and glaciation processes, but without causing significant changes to the dynamics of the already deep convective regime in the way that happens in the more susceptible shallow convection case.

> *l. 301: Please repeat which idealized studies this refers to.*

This refers back to the discussion in the penultimate paragraph of Section 4.2, and we have now repeated the citations here for clarity.

> *l. 301: I can see a number of potential problems with limited area studies, and I think that it is important to perform the type of study presented here. For example, applying a large scale forcing in a limited-area model means that in a strongly forced case, precipitation will more or less simply equal the large scale vapor large scale forcing. At the same time, it is not directly clear to me why phase changes due to precipitation formation delays should be less important in a strongly forced case, unless CCN are depleted by precipitation.*

As touched on above, the key argument here relates to the "tipping point" of glaciation – if a shallow convective cloud is in a state where it is not quite glaciating but an additional delay in warm rain production would cause it to do so, then it may be particularly susceptible to an aerosol-induced CDNC increase and this will lead to convective invigoration.

In strongly-forced deep convection, however, extra aerosol might modify the exact microphysical details inside the cloud (rebalancing warm vs cold rain for example) but providing glaciation still occurs, the impact on the overall convective dynamics may be minor. An exception would be if, in spite of delayed autoconversion, the delayed freezing of smaller droplets causes some clouds in the ensemble to lose buoyancy before they glaciate, leading to convective suppression – however there is little indication of that happening in these simulations. Where large-scale forcing is strong, it is able to maintain deep convection irrespective of the amount of aerosol.

> *l. 309: what does "via energetic and water-budget constraints" mean? Please be more specific.*

Having already expanded the discussion of this elsewhere in reference to fixed SSTs etc. we don't feel it's necessary to include this phrase here and have deleted it.

> *l. 310: Although invigoration might not change the time average amount of precipitation, it could still have an impact on precipitation intensity. Please discuss.*

We completely agree, and although the precipitation intensity is not the subject of this manuscript, we might expect to see some correlation between changes in the intensity distribution and the cloud-field distributions presented here. Although somewhat out of scope here, the baseline precipitation intensity distribution in CCFM was discussed in Labbouz et al. (2018), and an analysis of how this varies with aerosol perturbations could be a worthwhile topic for a future study. We have now noted this in the conclusions.

> *Fig 1: does this depend strongly on the model run?*

No: although the CCN distribution is of course modulated by changes in scavenging, this does not significantly change what is shown here. We have updated the caption to note that these are from the CCFMall_ari simulations, but that others look very similar.

> *Fig. 4: I think it would be good to explain the x-axis better. Perhaps you could mention the number of ensemble members in CCFM somewhere. I would like to obtain a rough idea about how the values on the x-axis come about without having to consult additional literature.*

A paragraph has been added to Section 2.2 explaining this, and a note to the captions of Figures 4 and 5. (There are 10 members in the ensemble in the CCFM configuration used here.)

> *Technical:*
> *l. 98: uses -> is diagnosed (and remove diagnostic in line 99)*

Changed as suggested.

> *l. 177: months' -> should it just be months without the apostrophe?*

Changed to "15 months of spin-up".

**Other corrections**

**line 64:** "considers attempts" $\longrightarrow$ "attempts".

**Figures 2, 3 and S1.** These had been incorrectly drawn using the _noari simulations to extract the CCFM response. We have corrected this to use the _ari simulations as indicated in Table 2, which provides for a logical progression of enabling successively more processes, is consistent with what is shown in the other figures, and does not change the conclusions drawn.

**Table 1.** We have updated this to show the correct set of simulations referred to in Table 2 (in particular, the large scale radiative effects are activated before the CCFM processes, not after). This was an oversight in updating from an earlier draft. The actual sets of differences listed in Table 2, however, were correct.

**References**

Labbouz, L., Kipling, Z., Stier, P., and Protat, A.: How Well Can We Represent the Spectrum of Convective Clouds in a Climate Model? Comparisons between Internal Parameterization Variables and Radar Observations, Journal of the Atmospheric Sciences, 75, 1509–1524, doi:10.1175/JAS-D-17-0191.1, 2018.

Lohmann, U.: Global anthropogenic aerosol effects on convective clouds in ECHAM5-HAM, Atmos. Chem. Phys., 8, 2115–2131, doi:https://doi.org/10.5194/acp-8-2115-2008, 2008.

[Figure]

Figure A1: Modified version of Figure 5 from the manuscript, showing how the regional cloud field response differs for the India region when it is restricted to land points only.

[Figure]

Figure A2: Modified version of Figure 6 from the manuscript, showing how the regional rain production response differs for the India region when it is restricted to land points only.

[Figure]

Figure A3: Modified version of Figure 7 from the manuscript, showing how the regional ice production response differs for the India region when it is restricted to land points only.